# Addition of *Lactobacillus fermentum* to Fermented Sea Buckthorn (*Hippophae rhamnoides* L.) Fruit Vinegar Significantly Improves Its Sour Taste

**DOI:** 10.3390/foods14071223

**Published:** 2025-03-31

**Authors:** Benhao Feng, Ruoqing Liu, Xiaolu Liu, Mingshan Lv, Shengchang Zhou, Ying Mu, Yao Zhao, Liang Wang

**Affiliations:** College of Life Sciences & Technology, Xinjiang University, Ürümqi 830046, China; fengbenhao2000@163.com (B.F.); liuruoqing6688@163.com (R.L.); liuxl3695@163.com (X.L.); lvmingshan6666@163.com (M.L.);

**Keywords:** fruit vinegar, characteristic metabolite, metabolomics, organic acid, flavor and texture

## Abstract

Fruit vinegar is typically produced through a two-stage deep liquid fermentation involving alcohol fermentation (*Saccharomyces cerevisiae*) and acetic acid fermentation (*Acetobacter pasteurianus*). In order to enhance the flavor and texture of sea buckthorn fruit vinegar, *Lactobacillus fermentum* was introduced into the alcoholic fermentation stage. At the end of fermentation, the total acid and acetic acid of sea buckthorn (*Hippophae rhamnoides* L.) fruit vinegar were both enhanced compared with sea buckthorn vinegar brewed through the traditional liquid fermentation method, and in terms of the main active ingredients, the total flavonoid content was slightly enhanced. Non-targeted metabolomics (LC-MS) was used to characterize the characteristic metabolite profiles during the fermentation process. A total of 55 differential metabolites, including organic acids, flavonoids, and amino acids, were identified, and the contents of citric acid, malic acid, and manganic acid, which are the sources of the irritating taste of sea buckthorn berry vinegar, were significantly reduced. In addition, the co-fermentation of *Lactobacillus fermentum* promoted both glycolysis and the TCA cycle and also led to a significant up-regulation of aromatic metabolites, such as ethyl acetate, ethyl lactate, and ethyl caproate. These results will provide new information on the dynamics of the characterized metabolites during the fermentation of sea buckthorn fruit vinegar.

## 1. Introduction

Sea buckthorn (*Hippophae rhamnoides* L.) is a deciduous shrub belonging to the genus sea buckthorn in the family *Elmaceae*. Sea buckthorn is recognized as having high nutritional and therapeutic value. More than 200 biologically active phytochemicals, including active substances, such as flavonoids, carotenoids, vitamins, sterols, and fatty acids, have been isolated and characterized from different parts of sea buckthorn [1]. The flavonoids, carotenoids, and vitamins in sea buckthorn berries have significant effects on human health, in addition to a range of health benefits, including anti-inflammatory, anticancer, antioxidant, immunomodulatory, cardioprotective, hepatoprotective, and antiaging properties, among others [2]. The high levels of vitamins C and E, as well as the high levels of total flavonoids and carotenoids, are the main reasons for the potent antioxidant activity, which helps in the scavenging of free radicals and the reduction of oxidative stress [3]. In addition, sea buckthorn is rich in fatty acids and amino acids, which are essential for human skin regeneration and wound healing through topical application [4]. Sea buckthorn has been used for thousands of years in Europe and Asia to treat various ailments ranging from skin disorders to cardiovascular diseases. Modern scientific research has confirmed many of the traditional uses of sea buckthorn, finding that sea buckthorn extracts can lower blood pressure, reduce lipid levels, modulate immune responses, and enhance liver function [5]. Recent studies have also highlighted the potential of sea buckthorn in controlling metabolic syndrome, including diabetes and obesity, due to its ability to improve insulin sensitivity and glucose metabolism [6].

Fruit vinegar is a vinegar product made from fruits through fermentation, which has received widespread attention due to its rich nutrients and multiple health benefits. In recent years, sea buckthorn fruit vinegar, as an emerging functional beverage, has gradually attracted the interest of researchers and consumers due to its unique flavor and multiple health benefits [7,8]. Sea buckthorn fruit vinegar not only retains the high nutritional value of sea buckthorn berries but also increases the content of its functional components through the fermentation process. Studies have shown that sea buckthorn berry vinegar has a variety of health benefits, such as antioxidant, anti-inflammatory, antimicrobial, hypolipidemic, and hypoglycemic effects [9]. Among them, flavonoids and vitamins in sea buckthorn berries were partially converted into more easily absorbed small molecule compounds during the fermentation process, which improved their antioxidant activity. In addition, the organic acids in sea buckthorn berry vinegar, such as lactic acid and acetic acid, not only give it a unique flavor but also have the effect of regulating intestinal microbiota and promoting digestion [10]. However, sea buckthorn berry vinegar made through single acetic acid fermentation usually has a strong sour and astringent flavor and pungency, which considerably limits its commercial attractiveness. The main compounds responsible for the poor sensory evaluation of sea buckthorn berry vinegar include organic acids, such as malic, citric, and quinic acids, high levels of fatty acids, as well as certain phenolic compounds [11]. These components, while contributing to the antioxidant capacity of sea buckthorn berry vinegar, pose a challenge to consumer acceptance due to their strong and sometimes unpleasant flavor. Efforts to improve the organoleptic properties of sea buckthorn berry vinegar have historically focused on blending it with other fruit juices, such as grapes or oranges, to balance its acidity and improve palatability [12] or employing food additives or sweeteners, amongst others, to enhance the taste. However, these methods, although effective to a certain extent, do not fully address the fundamental problem of the inherent sourness and odor of sea buckthorn berry vinegar, as well as its strong pungency.

Sea buckthorn fruit vinegar is traditionally produced through natural fermentation, but the natural fermentation process suffers from a long fermentation period and unstable flavor. Although liquid fermentation using *acetic acid bacteria* can shorten the fermentation time, its single fermenting strain still cannot completely solve the sour taste and odor problem of sea buckthorn fruit vinegar [13,14]. In addition, the diversity and abundance of microbial communities during fermentation may lead to unstable product quality [15], which makes the sour taste, odor, and irritation more intense, and these shortcomings limit market acceptance and application of sea buckthorn fruit vinegar.

To improve the flavor of sea buckthorn fruit vinegar, in recent years, researchers have tried to add *lactic acid bacteria* during the fermentation process. *Lactic acid bacteria* fermentation has the advantage of reducing acidity and improving flavor [16]. In previous studies on sea buckthorn wine and sea buckthorn enzymes, our laboratory found that screening of metabolites produced through the fermentation of sea buckthorn using suitable commercial *lactic acid bacteria* could significantly mask the bitter and astringent flavors of sea buckthorn [17]. In addition, metabolomics analysis revealed that the introduction of *Lactobacillus* altered the microbial community structure in the fermentation broth and reduced the production of undesirable metabolites. This improvement not only enhanced the flavor of sea buckthorn products but also maintained their high antioxidant activity and other nutritional values [18]. Sea buckthorn fruit vinegar fermented through traditional liquid fermentation with inoculation of *Saccharomyces cerevisiae* and *Acetobacter pasteurianus* was poor in taste and had a single flavor, and the dynamic changes of metabolites during fermentation of sea buckthorn fruit vinegar have not been reported. Here, *Lactobacillus fermentum* was introduced for lactic acid fermentation first, followed by alcoholic fermentation and acetic acid fermentation. Dynamic fermentation parameters of the sea buckthorn fruit vinegar fermentation system were collected to understand the effect of *Lactobacillus fermentum* on the taste and quality improvement of sea buckthorn fruit vinegar. The metabolic profiles during the fermentation of sea buckthorn fruit vinegar were investigated through LC-MS-based non-targeted metabolomics analysis and quantification of organic acids, total flavonoids, and total phenols, and key differential metabolic pathways were identified for each fermentation system. The information obtained here contributes to the understanding of the role of *Lactobacillus fermentum* in the fermentation of sea buckthorn fruit vinegar and deepens the understanding of the mechanisms of transformation of multiple metabolites.

## 2. Materials and Methods

### 2.1. Chemicals and Materials

The scientific name of sea buckthorn is *Hippophae rhamnoides* L. The use of sea buckthorn in this article indicates this. Sea buckthorn berries were supplied by Xinjiang Huize Food Co. in Xinjiang, China. Upon receipt, they were immediately stored at −18 °C in a refrigerator, model BCD-650WGHFD12STU1 (Qingdao, China). Pectinase, cellulase, and hemicellulase were purchased from Shanghai Yuanye Biotechnology Co. (Shanghai, China). Standards of amino acids, organic acids, and n-pentane (C_5_H_12_) were obtained from Sigma-Aldrich (St. Louis, MO, USA). All other chemicals and solvents were obtained from Sinopharm Chemical Reagent Co. (Shanghai, China). Rutin and gallic acid standards were obtained from Nanjing Jiancheng Co. (Nanjing, China).

### 2.2. Strains

The *Lactobacillus fermentum*, *Saccharomyces cerevisiae*, and *Acetobacter pasteurianus* used in this study were purchased from the China Industrial Microbial Strain Conservation and Management Center (CICC) (Beijing, China), and these strains are denoted as *Lactobacillus fermentum* F, *Saccharomyces cerevisiae* RV, and *Acetobacter pasteurianus* PAC, respectively. Strain details are described in the Appendix A.

### 2.3. Culture Medium

YPD medium (yeast extract 10 g/L, glucose 20 g/L, peptone 20 g/L) was used to activate *Saccharomyces cerevisiae* RV. MRS broth medium (peptone 10 g/L, yeast extract 5 g/L, ammonium citrate 2 g/L, glucose 20 g/L, dipotassium hydrogenphosphate 2 g/L, magnesium sulfate 0.58 g/L, manganese sulfate 0.28 g/L, beef extract 10 g/L, Tween-80 1 mL/L) was used for the activation of *Lactobacillus fermentum* F. AM medium (yeast extract 10 g/L, dextrose 20 g/L, ethanol 2.5%) was used for the activation of *Acetobacter pasteurianus* PAC.

### 2.4. Fermentation of Sea Buckthorn Vinegar

First, sea buckthorn was pre-treated to obtain sea buckthorn pulp. Sea buckthorn berries were cleaned and squeezed into a homogenized pulp using an electric juicer model MD-767 (Guangzhou, China). The homogenate was then placed in water at 90 °C for 15 min to inactivate the enzymes and microorganisms in the sea buckthorn itself. Then, the homogenate was cooled to room temperature, and 0.4% (*w*/*v*) pectinase, 0.2% (*w*/*v*) cellulase, 0.2% (*w*/*v*) hemicellulase, and 2.5% (*w*/*v*) sucrose were added and then enzymatically digested at 55 °C for 3 h. After enzymatic digestion, the obtained sea buckthorn pulp was inactivated at 80 °C for 30 min and then cooled to room temperature to obtain the treated sea buckthorn crude milk, which is denoted as LM here. LM was inoculated with *Lactobacillus fermentum* F (10^6^ CFU/mL, 1.5% *v*/*v*); the fermentation temperature was 37 °C, and the fermentation time was 20 h. Fermentation was carried out anaerobically in a thermostatic fermenter of the model BLBIO-1000SJA (Beijing, China) at 37 °C, and sea buckthorn lactic acid fermentation broth after *Lactobacillus fermentum* F fermentation was obtained, which is expressed here as LAB. Next, the LAB was adjusted to 14 °Brix by using sucrose and inoculated with *Saccharomyces cerevisiae* RV (10^6^ CFU/mL, 2% *v*/*v*). 2% *v*/*v*) at a fermentation temperature of 30 °C for 4 d. The fermentation was carried out anaerobically in a thermostatic fermenter at 30 °C to obtain sea buckthorn fruit wine, here called SC. The SC was then centrifuged to remove the cells and inoculated with *Acetobacter pasteurianus* PAC (10^6^ CFU/mL, 6% *v*/*v*) and subjected to acetic acid fermentation in a rotary oscillator incubator (180 rpm) at a temperature of 30 °C for 7 d. Acetic acid fermentation was completed, followed by centrifugation to remove the cells and retain the supernatant to obtain sea buckthorn berry vinegar fermented with the addition of *Lactobacillus fermentum* F, herein referred to as AC. For sea buckthorn berry vinegar formed through fermentation using the traditional liquid fermentation method, herein referred to as DC, the fermentation process was the same as that of AC, except that no *Lactobacillus fermentum* F fermentation was added. The LM was fermented for 4 d at 30 °C with *Saccharomyces cerevisiae* RV (10^6^ CFU/mL, 2% *v*/*v*) alone, resulting in sea buckthorn wine, herein referred to as SC0. After completion of the alcoholic fermentation, the subsequent acetic acid fermentation was performed as described above for AC. The centrifuges and shaking incubators used in the above operations were the MICROCEN2 centrifuge (Beijing, China) and the HNY-211BG rotary shaking incubator (Zhengzhou, China), respectively. All fermentations were carried out independently and in triplicate.

### 2.5. Analyses of Soluble Solids (TSS), Reducing Sugars, Ethanol, Acetic Acid, Organic Acids, Amino Acids, Total Acids, and Cell Growth (OD_600_)

Reducing sugar content was determined through the 3,5-dinitrosalicylic acid (DNS) method [19]. Soluble solids (TSS) were detected by using an Abbe refractometer model WAY-2WAJ (Shanghai, China) and expressed as Bailey’s saccharinity. Ethanol standard solutions were prepared and detected using HPLC, and standard curves were plotted to determine the ethanol content of the samples. The chromatographic column was a Phenomenex Hydro-RP C18 (250 mm × 4.6 mm, 4 μm); the detector wavelength was 215 nm, the column temperature was 30 °C, and the mobile phase was 0.2% metaphosphoric acid solution at a flow rate of 0.5 mL/min with a sample injection volume of 20 μL. The organic acids, such as acetic acid, oxalic acid, malic acid, and citric acid, were configured into the standard solution, which was analyzed using an Agilent 1200 HPLC (Agilent Technologies, model LC-1200, region Santa Clara, CA, USA.). Agilent 1200 high-performance liquid chromatograph and the detection method were the same as above. Total acids were determined through chemical titration [20]. The samples were analyzed using a Biochrom 30 amino acid autoanalyzer (Shanghai, China). The samples were diluted 10-fold with ultrapure water, and then 200 μL of the diluted solution was mixed with 800 μL of 2% salicylic acid and then left to stand for 30 min. It was then centrifuged for 10 min at 1300 rpm in an Eppendorf 5424R centrifuge (Hamburg, Germany). The supernatant was passed through a 0.22 μm aqueous filter membrane for instrumental detection. The growth of *Lactobacillus fermentum* F, *Saccharomyces cerevisiae* RV, and *Acetobacter pasteurianus* PAC during the fermentation process was detected using an Epoch2 enzyme marker from Bio Tek Instruments Co, Winooski, VT, USA.

### 2.6. Total Flavonoid (TFC) and Total Phenol (TPC) Substance Content

The total flavonoid (TFC) content was determined through the method of preparing rutin standards and calculated from the standard curve equation [21]. See Appendix A for detailed information on standard curves. Briefly, 1 mL of the sample was diluted 10-fold in a 10 mL volumetric flask, then 300 μL of 50 mg/mL NaNO_2_ was added and allowed to stand for 6 min. Then, 300 μL of 100 mg/mL Al(NO_3_)_3_ was added and left for another 6 min, and then 4 mL of 1 mol/L NaOH was added, and the volume was fixed with ethanol. Finally, absorbance was measured at 510 nm using a UV-3100PC model UV spectrophotometer (Shanghai, China). The total phenol (TPC) content was measured by using gallic acid as the standard, and the standard curve was plotted to measure the total phenol (TPC) content [22]. The procedure was as follows: dilute gallic acid standard solution with deionized water in different gradients, take 1 mL in a 10 mL stoppered cuvette, add 0.8 mL of folinol for 5 min, then add 1% sodium carbonate solution 1.5 mL, add water to 10 mL, shake well, and then put it in a dark place to react for 2 h. Finally, the absorbance was measured at 510 nm. The concentrations of both were obtained using a standard curve equation.

### 2.7. Metabolomics Analysis Using UPLC-MS/MS

All samples were thawed at 4 °C and vortexed for 30 s to mix well, and then an appropriate amount of liquid sample was placed in a 50 mL centrifuge tube and freeze-dried under vacuum at −80 °C. Then, 50 mg of the sample was weighed by using an electronic balance (MS105DM) (Mettler Toledo Instruments GmbH, sourced from Zurich, Switzerland), and 1200 μL of 70% methanol containing the internal standard extraction solution was added and vortexed with a steel bead for 15 min; the samples were then subsequently centrifuged at 12,000 rpm at 4 °C for 3 min; the supernatant was aspirated, and the sample was filtered through a microporous filter membrane (0.22 μm pore size) and stored in the injection bottle for UPLC-MS/MS analysis [23].

Three biological replicates were performed for each experiment. Chromatographic acquisition mobile phase A: ultrapure water (0.1% formic acid); mobile phase B: acetonitrile (0.1% formic acid). Instrument column temperature: 40 °C; flow rate: 0.4 mL/min; injection volume: 4 μL. Mass spectrometry analyses were carried out on a TripleTOF 6600+ Mass Spectrometer (SCIEX and the source is Shanghai, China), with ionization voltages of 5 kV and −4 kV in positive and negative modes, respectively. The spray gas and auxiliary heating gas were set to 50 and 60 arbitrary units, respectively. The capillary temperature was 325 °C. The analyzer scanned the mass range of *m*/*z* 25–1250 full scans at a mass resolution of 70,000. Data-dependent acquisition (DDA) MS/MS experiments were performed using HCD scans. The normalized collision energy was 30 eV. Dynamic exclusion was performed to remove some unwanted information from the MS/MS spectra. Centrifugation and freeze-drying were performed using an Eppendorf model 5424R centrifuge (Hamburg, Germany) and a LABCONCO CentriVap model freeze-drying concentrator (Kansas, MO, USA), respectively.

The raw mass spectrometry data were converted to mzXML format (a mass spectrometry data format mainly used for storing and analyzing mass spectrometry data of biological samples) using ProteoWizard (an open-source mass spectrometry data processing library), and the XCMS (an R package for metabolomics analysis) (XCMS.2.0) program was used for peak extraction, alignment, and retention time correction. Peaks with >50% missing rates in each group of samples were filtered, and blank values were filled with KNN (K-Nearest Neighbors) (a method to deal with missing values in data). Peak areas were corrected using the SVR (Support Vector Regression) method. The corrected and screened peaks were used for metabolite identification by searching the laboratory’s own database, integrating public libraries and prediction libraries, and using the MetDNA (Metabolite identification and Dysregulated Network Analysis) method. Finally, substances with a combined identification score of 0.5 or more and a QC (Quality Control) sample CV (Coefficient of Variation) value of less than 0.3 were extracted and then combined in positive and negative modes, and the collected data were exported to Excel for analytical and normalization purposes, using the peak area as a comparative measure on different scales.

### 2.8. Metabolic Pathway Analysis

Hypothesis testing and multiplicative analysis of variance were performed on the LC-MS results, and the Variable Importance in Projection (VIP) obtained based on the OPLS-DA (Orthogonal Partial Least Squares Discriminant Analysis) model (biological replicates ≥ 3) with a VIP value ≥ 1 could initially screen the metabolites for differences between species or tissues. For projection (VIP), a VIP value ≥ 1 can initially screen out metabolites that differ between species or tissues. At the same time, a *p*-value/FDR (False Discovery Rate) (biological repetition ≥ 2) or FC (Fold Change) value of the univariate analysis ≥ 2 was used to further screen out the differential metabolites, which were imported into the KEGG (Kyoto Encyclopedia of Genes and Genomes) database, and, for a given metabolite, it was imported into the KEGG (Kyoto Encyclopedia of Genes and Genomes) database. For a given list of metabolites, the pathways that were significantly enriched were identified using the *p*-value of the hypergeometric test [24]. The multichannel metabolic pathways were also mapped in conjunction with the analysis of changes in the content of differential metabolites.

### 2.9. Data Analysis

Metabolic profiles were analyzed using the SIMCA 16.0.2 software package (Sartorius Stedim Data Analytics AB, Umea, Sweden), which includes multivariate statistical analysis techniques, such as PCA (Principal Component Analysis) and OPLS-DA to differentiate between groups. PCA (Principal Component Analysis) was performed using the statistical function prcomp in R language programming (www.r-project.org, accessed on 20 January 2025). In the OPLS-DA analysis, a projected variable importance (VIP) score > 1.0 was considered to be a difference variable, and, in the *t*-test, variables with *p* < 0.05 were considered significant. Metabolites with VIP > 1 and *p* < 0.05 were selected as differential metabolites.

All experiments were performed in triplicate, and data were exported in triplicate for each sample for analysis. Analysis of variance (*t*-test) was then performed using SPSS software version 27.0 (SPSS-IBM Chicago, IL, USA). A significant difference was indicated at *p* < 0.05 and considered statistically significant.

## 3. Results and Discussion

### 3.1. Influence of Lactic Acid Levels on Yeast Fermentation and Acetic Acid Fermentation Processes

Prior to the fermentation of sea buckthorn fruit vinegar AC through addition of *Lactobacillus fermentum* F, the effects of lactic acid accumulation induced by *Lactobacillus fermentum* F on the growth and metabolic mechanisms of *Saccharomyces cerevisiae* RV and *Acetobacter pasteurianus* PAC in the subsequent fermentation phase were investigated (Figure 1). The results showed that the presence of lactic acid had a certain promotional effect on the growth and metabolic capacity of the two strains, and the biomass of *Saccharomyces cerevisiae* RV grew significantly and the alcohol production capacity increased dramatically at certain lactic acid levels. The alcohol yield reached 32.5 g/L after 48 h of fermentation (Figure 1D), which was much higher than that of 21.9 g/L at the level of 0 g/L lactic acid, which might be due to lactic acid production, which can adjust the pH value of the fermentation environment, thus inhibiting the growth of stray bacteria and ensuring that yeasts are in a dominant position during the fermentation process [25]. Meanwhile, *Acetobacter pasteurianus* PAC may utilize lactic acid as a carbon source and substrate for growth, resulting in an increased ability to ferment acetic acid. Under the ambient condition of 9 g/L of lactic acid, it was observed that *Acetobacter pasteurianus* PAC reached optimal growth and the highest acetic acid production, with a peak acetic acid yield of 13.8 g/L (Figure 1H). During the 24 h and 48 h time periods of fermentation, it was observed that the optimal growth intervals of both *Saccharomyces cerevisiae* RV and *Acetobacter pasteurianus* PAC were around the 9 g/L lactic acid level (Figure 1A,E), and the biomass and metabolite production of the bacteria at the 48 h period were significantly increased compared to the previous day (Figure 1B,F). It is noteworthy that *Saccharomyces cerevisiae* RV reached its maximum biomass at 9 g/L of lactic acid, and its alcohol production capacity was not much different from that at 6 g/L. This may be due to the high vigor and strong fermentation performance of *Saccharomyces cerevisiae* RV, which needs to use organic matter, such as glycogen and carbon, and nitrogen sources from raw materials for its growth and therefore took full advantage of its alcohol production capacity before reaching its maximum biomass. At low concentrations of lactic acid (0–3 g/L of lactic acid), both *Saccharomyces cerevisiae* RV and *Acetobacter pasteurianus* PAC grew slowly in terms of growth and ability to produce the corresponding metabolites (Figure 1C,G), but at 3–9 g/L of lactic acid, the growth and metabolism of these two bacteria were strong. And, after 12 g of lactic acid was reached, the ability of both bacteria was dramatically affected, and the fermentation power was lower than that of fermentation without lactic acid (Figure 1D,H). In conclusion, both *Saccharomyces cerevisiae* RV and *Acetobacter pasteurianus* PAC were able to utilize lactic acid for growth and metabolism, and although the optimal lactic acid levels of the two strains differed somewhat, maintaining the lactic acid level in a certain acidic environment (6–9 g) was significantly helpful for the fermentation process of sea buckthorn fruit vinegar.

### 3.2. Effect of Fermentation with Added Lactobacillus fermentum on Physicochemical Properties of Sea Buckthorn Fruit Vinegar

Many studies have shown that the optimal fermentation cycle of lactic acid bacteria is between 18 h and 30 h. After adding *Lactobacillus fermentum* F to start fermentation, the consumption of reducing sugar of sea buckthorn pulp decreased slowly, which was due to the consumption of *Saccharomyces cerevisiae* RV’s own growth and reproduction, and it had not yet started to utilize reducing sugar to produce ethanol until the eighth hour. Then, it began to decrease rapidly, which may be due to the fact that *Lactobacillus fermentum* F was still at the beginning of proliferation at the beginning stage of fermentation, and it utilized less reducing sugar. At each time stage after that, the reducing sugar consumption showed a 1.2–1.5-fold increase, and at 20 h, 70% of the reducing sugars of the sea buckthorn stock were consumed (Figure 2A). The titratable acid content of sea buckthorn berry vinegar continued to increase (Figure 2B), and its increasing trend almost corroborated with the decreasing trend of reducing sugars, which demonstrated that *Lactobacillus fermentum* F mainly metabolized and consumed reducing sugars from sea buckthorn to maintain its own life activities and produce a large amount of acids and flavor substances, which were mainly composed of lactic acid. The contents of total flavonoids and total phenols also increased steadily and reached the maximum values of 0.245 mg/mL and 0.363 mg/mL, respectively, at 20 h (Figure 2C,D), which may be attributed to the fact that most of the flavonoids in sea buckthorn existed in the form of unstable flavonoid glycosides, whereas fermentation of *Lactobacillus fermentum* F for de-glycosylation could convert flavonoid glycosides into the corresponding monomer substances, which significantly increased the bioavailability and stability of flavonoids [26]. The utilization of reducing sugars and the production of the corresponding metabolite alcohol by *Saccharomyces cerevisiae* RV was similar to that of *Lactobacillus fermentum* F (Figure 2E), but its metabolic cycle was longer, and the main metabolic activities started only from the second day, which was reflected in the consumption of reducing sugars. Furthermore, the growth of ethanol increased drastically from the second day onwards. On the fourth day of yeast fermentation, the ethanol content reached a maximum value of 83.6 g/L (Figure 2F), after which there was a partial decrease in the ethanol content. These results indicate that the optimal time period for the fermentation of sea buckthorn yeast is in the range of 4–5 days and that the decreasing ethanol was converted into other aromatic aldehydes, ketones, and esters, which resulted in a richer flavor of the sample’s mouthfeel. Compared with the malolactic fermentation stage, the trend of total polyphenols and total flavonoids in yeast fermentation showed a completely opposite state, and the contents of the two substances were in a decreasing state after a slight increase after the first day of fermentation (Figure 2G,H), which may be attributed to the fact that as fermentation proceeded, the alcohol content continued to increase, which led to a continuous leaching of flavonoids. Furthermore, during the middle and late stages of fermentation, the release of secondary metabolites by the yeast (pyruvic acid, acetaldehyde, etc.) may react with the flavonoids to produce some functional derivatives [27], resulting in a decrease in the content of both.

In the most important stage of acetic acid fermentation, the growth momentum of *Acetobacter pasteurianus* PAC of AC was also much stronger than that of DC, as shown by the 21.3% increase in the biomass OD_600_ value of *Acetobacter pasteurianus* PAC compared with that of the system without *Lactobacillus fermentum* F on the fifth day of fermentation (Figure 2I). And, the starting total acid amount of AC was 1.92 times higher than that of DC, which may be attributed to the fact that lactic acid accumulated during the fermentation stage of *Lactobacillus fermentum* F. The reducing sugar and alcohol consumption of AC was also significantly higher than that of DC from the fourth day onwards (Figure 2J), which indicated that the lactic acid produced in AC could be consumed as part of the energy metabolism of *Acetobacter pasteurianus* PAC after the rapid proliferation of *Acetobacter pasteurianus* PAC reached the optimal fermentation biomass in the first three days, thus indirectly enhancing the metabolism of ethanol respiratory chain and organic acids. It also promotes the conversion of ethanol to acetic acid and the production of various organic acids and aromatic substances. Notably, the decrease in TSS was significantly lower in AC than in DC (Figure 2I), which showed a higher abundance of substances produced through the metabolism of TSS by acetic acid under the influence of *Lactobacillus fermentum* F, yielding a wide range of water-soluble polysaccharides, organic acids, and vitamin-like metabolites. In addition, the total acid and acetic acid contents of AC were significantly higher than those of DC during fermentation and reached maximum values of 86.2 g/L and 73.3 g/L, respectively, on the sixth day of fermentation (Figure 2K), indicating that the presence of *Lactobacillus fermentum* F had a significant positive effect on the metabolic process of *Acetobacter pasteurianus* PAC, which showed a higher acetic acid production and accumulation rate compared with DC. During the fermentation process, the total flavonoid content of AC increased slightly, but there was no statistically significant difference; the total polyphenol content was in a decreasing state, and the total flavonoid and total polyphenol contents of AC were always higher than those of DC, with the total flavonoid and total polyphenol contents being 1.22 and 1.65 times higher than those of DC, respectively (Figure 2L). Furthermore, there was a statistically significant difference, which may be attributed to the fact that *Lactobacillus fermentum* F promotes acetic acid. This may be related to the fact that *Lactobacillus fermentum* F promotes the fermentation process and synthesizes abundant high-level metabolites with strong antioxidant activity, including ferulic acid and gallic acid [28], indicating a significant improvement in the antioxidant activity of AC vinegar enriched with *Lactobacillus fermentum* F.

### 3.3. Analysis of Metabolite Composition During Fermentation of Sea Buckthorn Fruit Vinegar

The non-volatile components of sea buckthorn crude milk LM, sea buckthorn lactic fermentation broth LAB, sea buckthorn wine SC and SC0, and sea buckthorn berry vinegar AC and DC were analyzed through LC-MS. By overlaying the total ion flow plots (TIC plots) analyzed through mass spectrometry detection in different QC samples, the results showed that the curves of the total ion flow for metabolite detection had a high degree of overlap, i.e., the retention time and the peak intensities were the same, indicating that mass spectrometry had good signal stability when detecting the same samples at different times. The signal stability is good when detecting the same sample at different times. The data of QC samples were also analyzed using Pearson correlation analysis to ensure stable data quality. A total of 4836 metabolites were detected in different metabolome groups, covering 12 major classes of primary metabolites (Appendix A). The five categories of organic acids, amino acids and their derivatives, flavonoids, phenolic acids, and nucleotides and their derivatives were the most abundant, and about 196 flavonoids and 145 phenolic acid metabolites were detected. The joint proportion of organic acids and amino acids reached 52% of all metabolites, covering 621 organic acids and 1924 amino acid derivatives (Figure 3A,D). This indicates that organic acids and flavor-presenting amino acids have a large impact on the taste and flavor of sea buckthorn fruit vinegar, and they have an important influence on the samples [29,30,31]. To investigate the effect of the addition of *Lactobacillus fermentum* F fermentation on whether the taste of sea buckthorn fruit vinegar is improved, a five-component PCA analysis of the metabolite dataset was performed according to the fermentation stage, which showed that the data from the five samples were well-separated and had significant clustering behavior. The first two principal components (PC1 and PC2) were identified, which accounted for 39.08% and 20.56% of the variance, respectively. PCA score plots (Figure 3C) showed the trend of intergroup separation of sea buckthorn berry vinegar at different stages of fermentation, and the separation of the main components of sea buckthorn pulp, AC vinegar, and DC vinegar was obvious.

To further analyze the changes in metabolite composition at each fermentation stage, we identified 70 (VIP ≥ 1, *p* ≤ 0.05) metabolites from sea buckthorn stock, sea buckthorn wine, and sea buckthorn vinegar as differential compounds distinguishing between the LM, LAB, SC, SC0, AC, and DC samples (Figure 3B), which covered 20 organic acids, 15 amino acids and their derivatives, six alcohol ester metabolites, five kinds of lipids, nine kinds of phenolic acids, and seven kinds of flavonoids (Appendix A). It can be seen that organic acids, amino acids, and phenolic acids are the fundamental constituent substances that cause the differences in each sample [32]; in addition, with fermentation, the metabolite types and contents of sea buckthorn berry vinegar changed significantly, and its metabolite level became more and more abundant. The differential metabolites in each type of sample were analyzed through grouping of primary metabolites (Figure 3E), and the results showed that compared with LM sea buckthorn stock, sea buckthorn wine SC and sea buckthorn berry vinegar AC exhibited higher levels of aromatic alcohols, ester metabolites, as well as organic acid and amino acid metabolites, and it is worth noting that the levels of their lipid metabolite substances were significantly reduced, especially 12-hydroxystearic acid, ethyl linoleate, and iso palmitic acid, indicating that lactic acid fermentation can effectively remove the high level of lipid components in sea buckthorn pulp, significantly promote the metabolism of lipid substances, and improve the bad flavor and greasy taste of sea buckthorn. The staged fermentation of fruit vinegar using *Lactobacillus fermentum* F, *Saccharomyces cerevisiae* RV, and *Acetobacter pasteurianus* PAC resulted in significantly higher levels of organic acids, amino acids, and flavonoid metabolites than those in the DC group, as well as a greatly altered composition of organic acid species. In conclusion, with the participation of *Lactobacillus fermentum* F, the AC system exhibited higher levels of organic acids and amino acids and lower levels of liposomal lineages, as well as richer levels of aromatic metabolites, compared with the raw sea buckthorn juice and DC vinegar.

### 3.4. Changes in Differential Metabolites in Fermentation Systems with the Addition of Lactobacillus fermentum F

The poor taste of sea buckthorn berry vinegar is mainly due to the presence of various irritating organic acids and phenolic acid compounds and the composition of a high lipid system. To investigate the effect of the addition of *Lactobacillus fermentum* F fermentation on the taste of sea buckthorn berry vinegar as well as changes in the metabolite composition, supervised chemometric analyses of OPLS-DA were established using the AC, LM, LAB, SC, and DC datasets (Figure 4A). Different groupings were achieved according to the fermentation process (Appendix A), and 55 metabolites were identified as the metabolites that differentiated between sea buckthorn raw milk (LM), lactic acid fermentation broth (LAB), sea buckthorn wine (SC), and sea buckthorn fruit vinegar (AC and DC) samples (Figure 4B).

These differential metabolites included various compounds covering organic acids, alcohols, esters, amino acids, flavonoids, alkaloids, and other related substances [33]. The relative contents of acetic acid, lactic acid, and tartaric acid were higher in the AC, followed by several esters, such as ethyl lactate and ethyl acetate, and a small fraction of phenolic acids, such as ferulic and vanillic acids. At the same time, lipids, such as oleic acid and stearic acid, were at the lowest level. Some lipid metabolites were more abundant in the unfermented sea buckthorn pulp, including palmitoleic acid, linoleic acid, and linolenic acid, whose levels decreased rapidly with fermentation, in addition to the increasing abundance of amino acids [34]. Compared with the DC system, the three types of organic acids, lactic acid, acetic acid, and succinic acid, showed a significant upward trend in the AC system, indicating that *Lactobacillus fermentum* F had a positive influence on the metabolism of organic acids during the fermentation process. Notably, four types of organic acids, including oxalic acid, citric acid, malic acid, and manganic acid, were significantly reduced in the AC vinegar system, which may be attributed to the fact that the accumulation of lactic acid through *Lactobacillus fermentum* F fermentation enhances the TCA cycle, metabolizing organic acids and supplying energy, and promotes the metabolism of ethanol and acetic acid [35,36].

In counting and comparing the metabolites of each fermentation stage (Figure 4C), we found that the metabolites common to the five fermentation systems were 244, most of which were amino acids and their derivatives, and 208 metabolites were unique to AC, mostly alcohol esters, organic acids, and flavonoids, which was significantly increased in both types and numbers compared to the other systems. This indicated that the abundance of metabolites in sea buckthorn fruit vinegar was significantly enhanced with the participation of the *Lactobacillus fermentum* F fermentation system compared with the DC traditional process fruit vinegar. The screened metabolites were statistically analyzed, and grouped differential metabolite volcano plots were drawn to compare the metabolic differences among individual samples (Figure 4D,E). The results showed that there were significant differences in metabolites between individual metabolic groups, and the top three significantly different metabolites were lactic acid, malic acid, and ethyl lactate in the comparison of the AC and DC systems, and the most significant metabolite was acetic acid in comparison to sea buckthorn stock LM. Notably, in the DC system, there was a significant difference in citric acid levels when comparing sea buckthorn stock. In the data of relative content changes of differential metabolites, we found that 38 metabolites were up-regulated in the AC vs. DC system, with lactic acid, ethyl acetate, and tartaric acid being significantly up-regulated, and 15 metabolites were down-regulated, with butyric acid, stearic acid, and oleic acid, which have undesirable flavors, being significantly down-regulated. This indicates a significant change in the composition of stimulants in sea buckthorn berry vinegar, which may significantly improve its flavor.

### 3.5. Analysis of Key Metabolic Pathways in Sea Buckthorn Vinegar

The metabolism of organic acids and amino acids is closely related to the differential metabolic pathways of sea buckthorn berry vinegar [37,38,39]. To elucidate the specific metabolite changes associated with these metabolic pathways during the fermentation of berry vinegar as well as the reasons for the change in the quality of the berry vinegar, we identified metabolite analyses through LC-MS-based metabolomics and expressed differences in the metabolite enrichment between the various stages of fermentation (Appendix A), and we identified the metabolic pathways that exhibited differential expression of the AC, DC, and LM systems, the main metabolic pathways that were differentially expressed (Figure 5). The results showed that the addition of the *Lactobacillus fermentum* F fermentation process enhanced glycolysis, the TCA cycle, ethanol metabolism, citric acid metabolism, manganic acid metabolism, and oxaloacetate metabolism. In addition, it had a positive effect on glutamate and aspartate metabolism and the synthesis of flavonoids, such as kaempferol, quercetin, etc. The TCA cycle serves as a key link between various pathways involved in the anabolic and catabolic metabolism of the substances and regulates the electron flux in the respiratory chain of the microorganisms to produce a large amount of ATP to supply energy for the metabolic processes [26,40]. Malic acid, the main organic acid in the fermentation phase, was reduced by 82% and 61% in AC and DC spoonfuls of vinegar, respectively, while there was no significant rise in citric acid in AC compared to DC. In addition, it is worth noting that the metabolism of many intermediates varies greatly in the TCA cycle; specifically, fumaric acid, succinic acid, and oxaloacetic acid were more abundant in AC, whereas malic acid was more abundant in DC. These changes suggest that abundant malic acid was utilized to a greater extent as a carbon source with the participation of the *Lactobacillus fermentum* F fermentation system, which resulted not only in a greater abundance of organic acids in AC vinegar but also in a reduction of irritant acids. In manganic acid metabolism, manganic acid in AC was significantly reduced, and it was converted to aromatic amino acids, such as phenylalanine, tyrosine, and tryptophan, in large quantities. Tyrosine was also involved in the metabolic pathway of leucine and valine, followed by the conversion to caffeic acid and ferulic acid, which have significant antibacterial and antioxidant properties. The glutamic acid in sea buckthorn, as an acidic flavor-presenting amino acid, was significantly reduced through fermentation and converted into amino acid derivatives, such as succinyl-L-arginine. Regarding the metabolism of pyruvate, the lactic acid, acetic acid, and acetyl coenzyme A accumulated during AC fermentation were significantly increased, and, accordingly, the precursor substance pyruvate was significantly depleted as a metabolic substrate, part of which was used for secondary metabolism, and most of which was used to generate ATP through phosphorylation to supply energy for the fermentation process. Higher energy was more favorable for the winemaking and vinegar-making process compared to the traditional fermented vinegar DC, where the addition of *Lactobacillus fermentum* F fermentation promoted pyruvate metabolism as well as enhanced utilization of stimulating acids, such as malic and citric acids, which may explain the higher levels of ethanol and acetic acid yields in AC. In the metabolism of flavonoids, a significant decrease in the relative content of kaempferol and quercetin was observed, which may be attributed to the fact that they are based on 2-phenyl chromogranin, which reacts with hydroxyl radicals during yeast fermentation, resulting in a decrease in the concentration of metabolites [41]. Nevertheless, flavonoid metabolites identified in the individual fermentation systems showed a wide range of content and forms of presence, and the complex biochemical processes underlying them, which are linked to microbial fermentation, might be the complex biochemical processes associated with microbial fermentation. There is a great need to investigate the underlying biochemical mechanisms through which microbial physiological activities affect the formation of functional substances, such as flavonoids [33,42].

In summary, the differences in metabolic pathways and the significant differences in metabolite contents between AC and DC indicated that the presence of lactic acid in AC vinegar promoted the energy conversion mechanism of microorganisms, with higher energy production, which led to an enhanced metabolic flux of stimulating acids, such as malic, citric, and oxalic acids, and accelerated the process of yeast and acetic acid fermentation. In addition, there was a significant increase in the content of soft acids, such as lactic acid, and the composition of phenolic acids, such as gallic acid, was significantly increased. The composition of phenolic substances was changed, resulting in a great improvement in the taste of AC vinegar and a greater abundance of secondary metabolites. Sea buckthorn fruit vinegar fermented with added *Lactobacillus fermentum* F may have enhanced health benefits, e.g., caffeic acid has antioxidant and anti-inflammatory effects and helps to reduce inflammatory responses, and nicotinic acid and ferulic acid reduce the progression of cardiovascular and coronary heart disease, as well as atherosclerosis and other clinical events [39,43]. In conclusion, the addition of *Lactobacillus fermentum* F promotes the metabolic process of stimulating acids, such as mangiferic acid, citric acid, malic acid, etc., which are converted into various aromatic amino acids and secondary metabolites while also generating a large amount of energy, increasing the rate of glycolysis and the TCA cycle. This benefits from the accumulation of lactic acid in *Lactobacillus fermentum* F fermentation in addition to promoting the process of yeast and acetic acid fermentation and a wide range of phenolic acids and functionalities, such as kaempferol, quercetin, isorhamnetin, and other functional substances, at a richer level.

## 4. Conclusions

This study comprehensively describes the significant improvement of the bad taste of sea buckthorn berry vinegar and facilitation of the fermentation process through the addition of *Lactobacillus fermentum* F assisted fermentation. *Acetobacter pasteurianus* PAC benefited from the assisted fermentation with *Lactobacillus fermentum* F, which enhanced the TCA cycle and energy metabolism and consequently improved ethanol conversion and acetic acid synthesis efficiency. A total of 4836 non-volatile metabolites were qualitatively identified through LC-MS, of which 55 differential metabolites screened had a decisive impact on the overall mouthfeel and flavor of sea buckthorn berry vinegar AC. Pungent acids and lipids were the main factors contributing to the unpleasant taste in the mouth of sea buckthorn fruit vinegar. Then, OPLS-DA analysis revealed that butyric acid, stearic acid, citric acid, and oleic acid were the main components contributing to the unpleasant taste, and these substances were significantly down-regulated in the AC fermentation system. In addition, the fermentation system containing *Lactobacillus fermentum* F showed a metabolic pathway for the conversion of mangiferic acid to aromatic amino acids, and it accumulated a richer amount of ethyl lactate, ethyl acetate, and ethyl caproate, which enhanced the sea buckthorn palatability of fruit vinegar AC. In the future, there is still a wide range of research space and prospects for studying the mechanism of interaction between microorganisms and metabolites and elucidating the basic mechanisms of the metabolic pathways of a wider range of functional metabolites.

## Figures and Tables

**Figure 1 foods-14-01223-f001:**
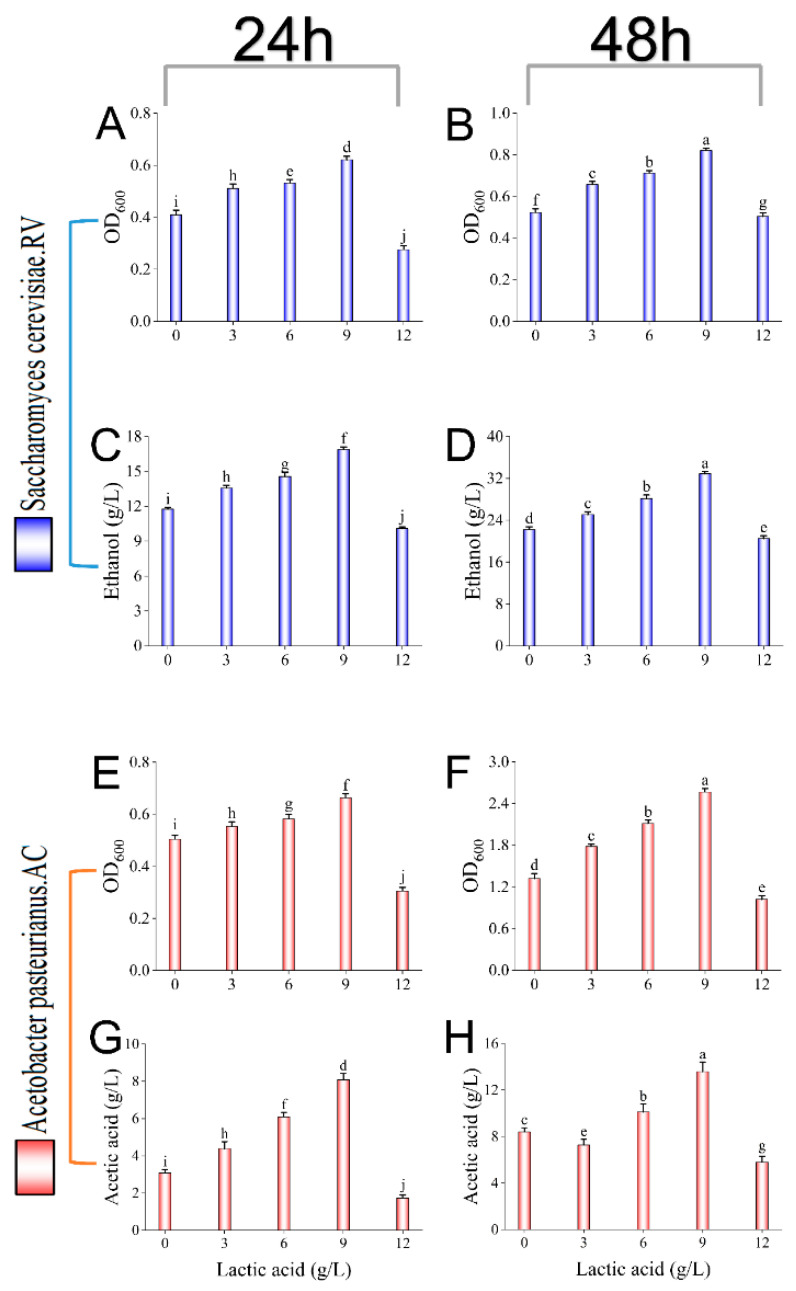
Effect of different lactic acid levels on growth and metabolite production of *Saccharomyces cerevisiae* RV and *Acetobacter pasteurianus* PAC. *Saccharomyces cerevisiae* RV: 24 h of biomass (**A**); 48 h of biomass (**B**); 24 h of ethanol production (**C**); 48 h of ethanol production (**D**). *Acetobacter pasteurianus* PAC: 24 h of biomass (**E**); 48 h of biomass (**F**); 24 h of acetic acid production (**G**); 48 h of acetic acid production (**H**). Different letters in the same graph indicate statistically significant differences in the results (*p* < 0.05).

**Figure 2 foods-14-01223-f002:**
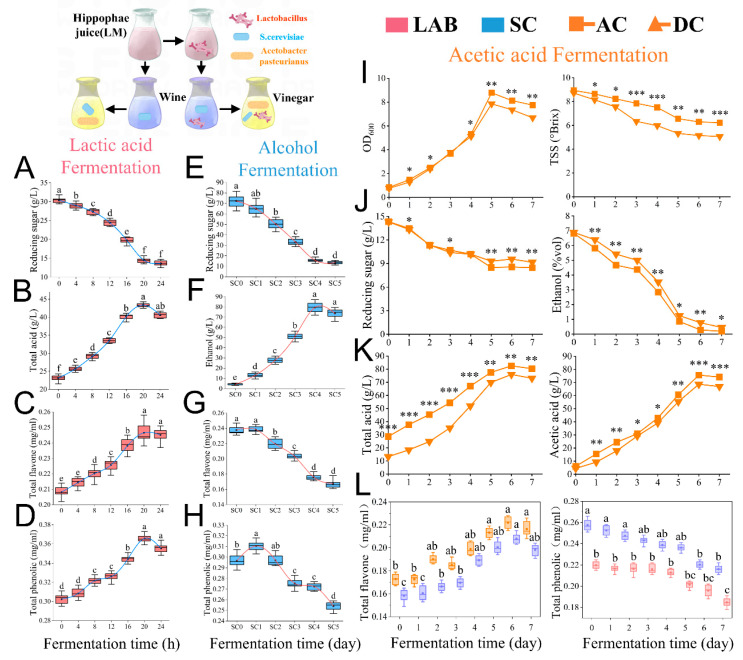
Changes in some physicochemical properties and main active components of sea buckthorn fruit vinegar during fermentation. Lactic acid fermentation stage: reducing sugar (**A**); total acid (**B**); total flavonoids (**C**); total phenols (**D**). Alcoholic fermentation stage: reducing sugar (**E**); ethanol (**F**); total flavonoids (**G**); total phenols (**H**). Acetic acid fermentation stage: biomass (expressed as OD_600_) and soluble solids (TSS) (**I**); reducing sugar and ethanol (**J**); total acid and acetic acid (**K**); total flavonoids and total phenols (**L**). Note: different letters in the same figure indicate statistically significant differences in results (*p* < 0.05); *t*-test, * *p* < 0.05, ** *p* < 0.01, *** *p* < 0.001.

**Figure 3 foods-14-01223-f003:**
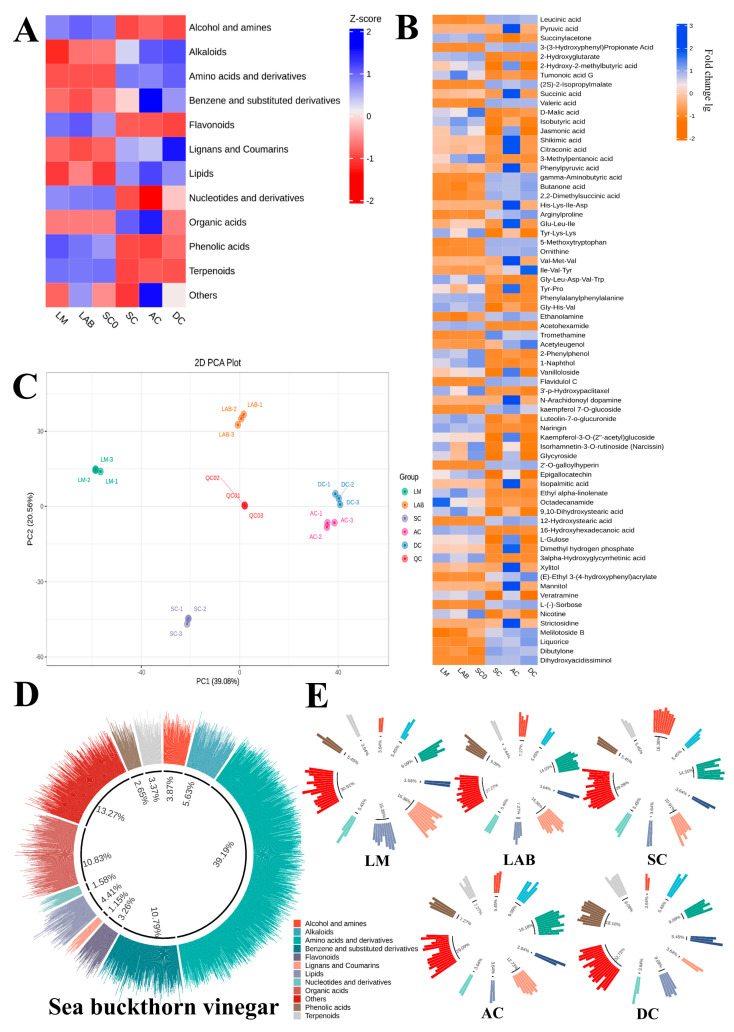
Metabolite profiles of sea buckthorn stock, sea buckthorn wine, and sea buckthorn fruit vinegar with and without lactobacillus fermentation. Substance level classification heat map (**A**); differential metabolite heat map (**B**); PCA score map (**C**); total metabolite substance classification map (**D**); differential metabolite level classification map (**E**).

**Figure 4 foods-14-01223-f004:**
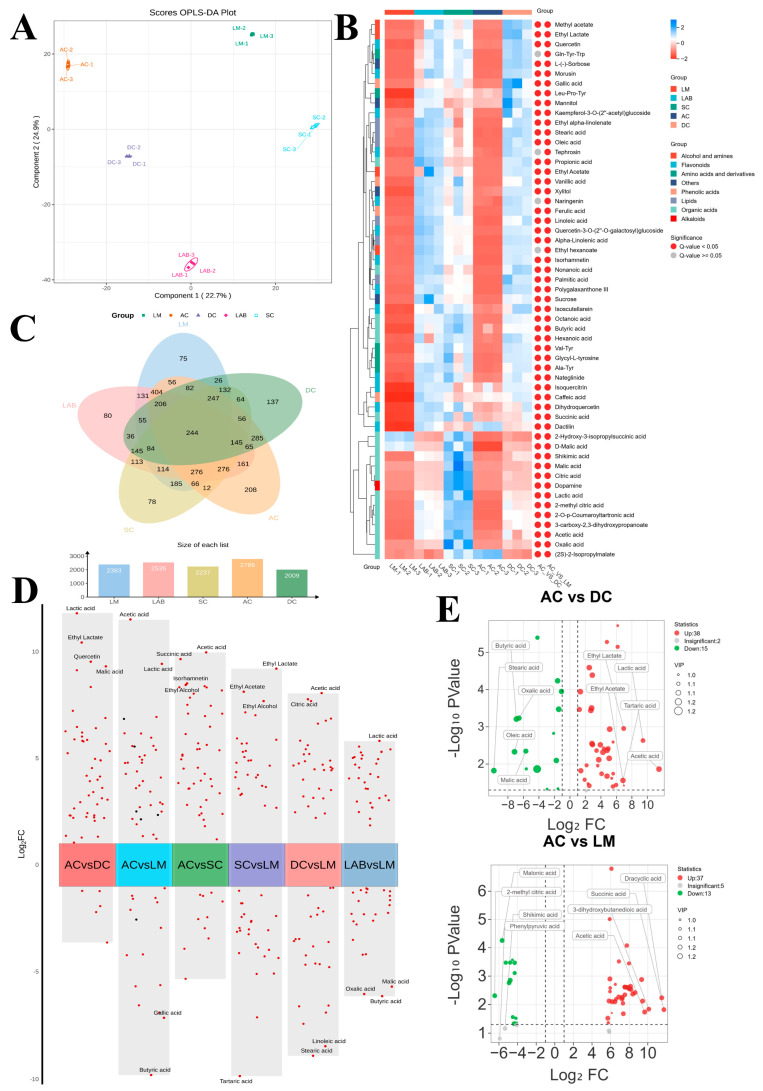
Metabolite composition analysis of the five fermentation groups: Orthogonal Partial Least Squares Discriminant Analysis (OPLS-DA) scoring plots based on positive ionization LC-MS data (**A**); heatmap of differential metabolites based on VIP ≥ 1 screening (**B**); Wayne plots (**C**); sub-grouped differential metabolite volcano plots (**D**); differential metabolite volcano plots (**E**).

**Figure 5 foods-14-01223-f005:**
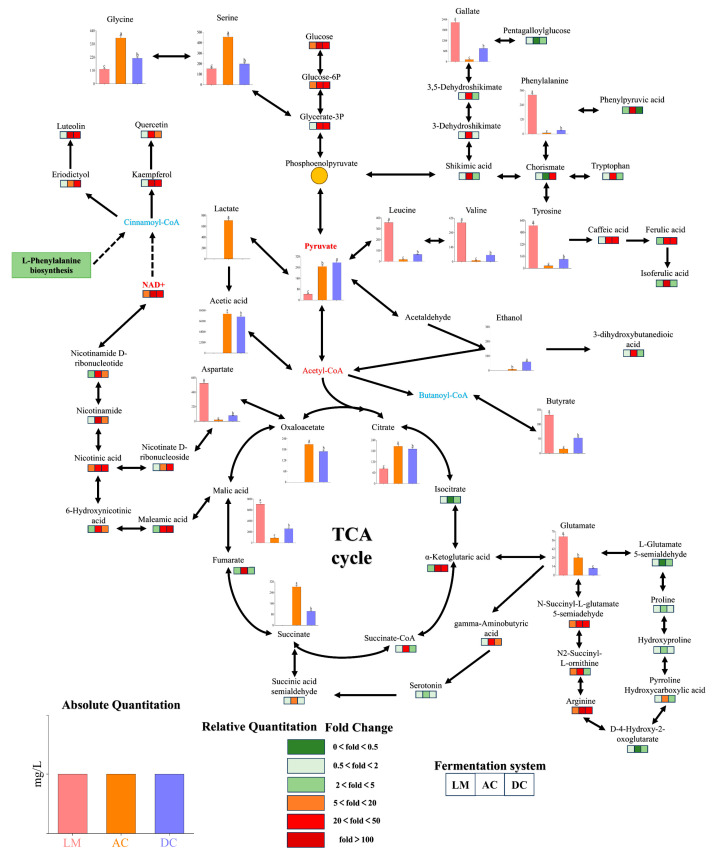
TCA cycle, glycolysis, manganic acid metabolism, citric acid metabolism, and anabolic pathways of acetic acid and flavonoids during acetic acid fermentation of sea buckthorn berry vinegar. Different letters represent significant differences between samples (*p* < 0.05).

## Data Availability

The original contributions presented in the study are included in the article/Appendix A, further inquiries can be directed to the corresponding author.

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
