# Peer review of "Addition of Lactobacillus fermentum to Fermented Sea Buckthorn (Hippophae rhamnoides L.) Fruit Vinegar Significantly Improves Its Sour Taste"

_foods, 2025, doi:10.3390/foods14071223_

Round 1
Reviewer 1 Report (Previous Reviewer 2)
Comments and Suggestions for Authors
Dear Authors
I acknowledge the significant change and improvement to the manuscript, which reflects the professionalism of the working group. However, I can send some additional comments to polish the final details of the manuscript.
Abstract
Please review the abstract carefully. The authors' guidelines state that the maximum word count is 200 words.
Keywords
I recommend that the keywords be different from those that appear in the title.
Materials and Methods
Pag. 3-5. Lines 140, 162, and 213. Use abbreviations appropriately. For example, replace "minutes" to "min." Change “days” to “d”. Change “r/min” to “rpm”. Please review the rest of the document.
Results and Discussion
Figures. Please check the resolution of all the figures carefully. They become less sharp as you zoom in.
References
Review the format of the references in detail. There are many inconsistencies.
Pag. 17. Line 63. For example, the author’s guidelines do not allow the use of "et al" or "&". Review the entire list.
In general, the name of journals should be abbreviated. Check that scientific names appear in italics. Review the entire list.
Comments on the Quality of English LanguageThe quality of the language is acceptable. I don't rate the quality of English.
Author Response
Please see the attachment.

Reviewer 2 Report (Previous Reviewer 3)
Comments and Suggestions for Authors
Thanks to my suggestions, the authors have made profound changes to the manuscript, which have been resolved and have greatly improved the manuscript.
Finally, the authors should check the following:
Line 156: for After
Line 195: NaNO2
Line 196: Al(NO3)3
Line 197: 6 min. finally
References:
Check references 27 and 37.
The authors should include in their manuscript the following paper, recently published in FOODS, because it deals with acetic acid microorganisms and their metabolites:
Román-Camacho, J.J.; Santos-Dueñas, I.M.; García-García, I.; García-Martínez, T.; Peinado, R.A.; Mauricio, J.C. Correlating Microbial Dynamics with Key Metabolomic Profiles in Three Submerged Culture-Produced Vinegars. Foods 2025, 14, 56. https://doi.org/ 10.3390/foods14010056.
Author Response
Please see the attachment.

This manuscript is a resubmission of an earlier submission. The following is a list of the peer review reports and author responses from that submission.
Round 1
Reviewer 1 Report
Comments and Suggestions for Authors
In the manuscript “Addition of Lactobacillus fermentum fermented sea buckthorn fruit vinegar significantly improves its sour taste” authors aim to study how to improve the taste of sea buckthorn fruit vinegar by the addition of a lactobacillus strain.
The topic is interesting, but the manuscript needs a deep improvement. It is not suitable for publication in its present form, and methods should be improved since it is not reproducible as it is written, because many data were missed of is contradictory as the name of the species.
Some comments to improve the manuscript:
Title:
The title previews the results.
The scientific name of 'sea buckthorn' should be included in the title.
Lactobacillus fermentum should be written in italic in the title and in the whole manuscript, as all the name of all the microorganisms used.
The strain of Lactobacillus fermentum used should be stated in the title.
Abstract:
When is the strain added to the vinegar? How is vinegar produced? Is it a spontaneous fermentation or is an acetic acid bacteria strain added?
The statement: 'the production of lactic acid by Lactobacillus fermentum enhanced glycolysis and pyruvic acid metabolism during the fermentation process' needs justification.
Lines 21-14: the phrase “In addition, the addition of Lactobacillus fermentum fermentation led to a significant up-regulation of aromatic metabolites such as ethyl acetate, ethyl lactate, and ethyl caproate, suggesting that the addition of Lactobacillus fermentum fermentum could improve the sour taste of traditional fermented sea buckthorn vinegar” is redundant. Please check the grammar of the sentence.
INTRODUCTION:
Scientific names of plant species should be written in italics, as well as the names of the family, class, and any other taxonomic level when written in Latin.
It is suggested that the health benefits listed in the introduction be limited to those obtained from consuming the plant or the vinegar, and not to mention those related to skin wound treatment through topical application, for example. Or, if mentioned, it should be specified that these benefits do not come from consuming the product orally, but from applying it topically (lines 39-40)
Lines 58-60: authors claim: In addition, the organic acids in sea buckthorn berry vinegar, such as lactic acid and acetic acid, not only give it a unique flavour but also have the effect of regulating intestinal flora and promoting digestion [10].”
But in reference 10 (Wang, Y. , Shan, Q. , Jia, Y. , Wu, T. , Zhang, J. , & Shan, L. . (2024). Ultrasound-assisted acidic natural deep eutectic 606 solvent as a new strategy for extracting seabuckthorn leaf phenolics: process optimization, compositional identification, and 607 metabolic enzyme inhibition capacity. Food & Bioprocess Technology, 17) authors do not study intestinal flora regulation. So, please check to put the appropriate references, her and in all the manuscript.
Authors claim” In previous studies on sea buckthorn wine and sea buckthorn enzymes, our laboratory found that the metabolites produced by fermentation of sea buckthorn by screened suitable commercial lactic acid bacteria could significantly mask the bitter and astringent flavours of sea buckthorn. In addition, the introduction of Lactobacillus was found to alter the microbial community structure in the fermentation broth and reduce the production of undesirable metabolites by macroeconomics analysis [17]. “
But reference 17 (Liu, J. , Huang, H. , Han, L. , Zhang, D. , Wang, P. , & Xie, X. , et al. (2024). Study on the mechanism of natural polysaccharides on the deastringent effect of triphala extract. Food Chemistry(May 30), 441) do not deal with that.
The same for reference 18. Please check the references.
Moreover: what means “macroeconomic analysis”?
Line 95: which is the “uniform design mixture”?
Lines 95-100 are not enough for the justification of the selection of the strain, strain origin, study design. Please deepen the justification of the study. Please put the name of the bacteria in italics.
Materials and methods
Lines 111-114: please put the name f the MO in italics. Which name of strains were used? Where them characterized or identified somewhere else? Please put the proper references to them in the reference list. If the strains are using here for the first time: how were then identified? Why where the strains selected? Why did authors name a Bacillus pasteurus aceticus that is not named in other parts of the manuscript?
Lines 116-124: why were used 3 different mediums for Lb grow? Why is named a culture medium for Acetobacter which is not listed in lines 111-114.
Line 123: Acetobacter pasteurii is named here for the first time. Which is its provenance? Name of the strain? Please add it tho the list of strains used (lines 111-114)
Line 130: which sugar was added?
Line 131: at 80°C no sterilization occurs.
Line 132: what is the LM?
Lines 132-133: Another new specie appears: “Lactobacillus fermentum was then inoculated with Lactobacillus fermentans, and the fermentation temperature was 37°C for 20 h to obtain the LAB.” Please check this information. Moreover: What means “LAB”?
Line 134: how was performed the “alcoholic fermentation stage”? dis authors used yeasts and bacteria?
Line 137: A new specie appears now: Bartonella acetogenes in a new media: “seed culture”. Please complete the information in the proper section of Materials and Methods.
Lines 138-141: what are SC, AC and DC
Line 141: Saccharomyces cere visite ?
Line 164: this specie appears here for the firs time: Acinetobacter pasteurianus. Which is its provenance? Name of the strain? Please add it the list of strains used (lines 111-114)
Line 167: what is TFC content? Concentration or content?
Line 172: “TPC The total phenol content”?
Lines 167-178: it is not clear which were the stard used for total phenol content. Rutin, foraminol or gallic acid?
Lines 189-191: please check grammar or punctuation.
Line 200: what means “(Ref)?)
Line 204: which is the SRV method? And the met DNA method? Please give a reference.
Line 208: what is a “QC sample CV value”. Please put the name complete of the initials. The same for lines 212-220.
Lines 167-168: authors claim “The TFC content was determined by preparing rutin standards and calculated from the standard curve equation (Figure 1)” but Figure 1 do not show TFC.
Results and Discussion:
In general: the experimental design it is not properly explained. It is not clear if authors ferment the vinegar with Lb first and then with another MO (yeast? Acetobacter? Which specie of Acetobacter? Since two species are named). Please revise the whole manuscript in order to avoid contradictions.
Lines 240-244: this referee do not understand the meaning of this phrase (Lb fermentum was used or not at this stage?):
“Prior to the fermentation of sea buckthorn fruit vinegar through the addition of Lactobacillus fermentum, the effects of lactic acid accumulation produced during Lactobacillus fermentum fermentation on the growth and metabolic mechanisms of Saccharomyces cerevisiae and Acetobacter pasteurianus in the subsequent fermentation
phase were investigated”.
Figure 1: a new specie a bacteria appears here: Acetobacter baumannii AC. Which is its provenance? Name of the strain? Please add it the list of strains used (lines 111-114)
Line 373: Which are Figure A and B?
Line 267 and 268: which “both bacteria”?
Line 444: how is possible this number: “244, 90%”
Line 547: Acetobacter pasteurianus appears in the conclusions for the first time. Which is its provenance? Name of the strain? Please add it the list of strains used (lines 111-114).
Comments on the Quality of English Language
Language must be improved.
Reviewer 2 Report
Comments and Suggestions for Authors
Dear Authors
In this research was planned to adopt the screened Lactobacillus fermentum to assist the fermentation of sea buckthorn fruit vinegar and to explore the effect of the addition of Lactobacillus fermentum on the improvement of the astringent flavor texture and quality of sea buckthorn fruit vinegar. However, I can send some observations to improve the manuscript:
Abstract
I recommend using italics in the scientific name of the microorganism (review this detail throughout the document).
Materials and Methods
Pag. 3. Line 104. Freezer or refrigerator? Add name, model, and country of origin of equipment.
Pag. 3. Lines 111-112. Names of microorganisms in italics.
Pag. 3. Line 127. Add name, model, and country of origin of electric juicer.
Pag. 3. Line 127. I recommend making abbreviations consistent throughout the document. For example, replace "minutes" to "min."
Pag. 3. Lines 137-140. Add name, model, and country of origin of centrifuge and rotatory oscillator.
Pag. 4. Lines 148-149. I recommend adding references for the DNS method and soluble solids.
Pag. 4. Line 168. This line cites Figure 1 and mentions the preparation of the curves. However, Figure 1 does not show this information. Please review this information carefully.
Pag. 4. Lines 167-179. I recommend adding references for the TFC and TPC methods.
Pag. 4. Lines 172 and 178. Add name, model, and country of origin of spectrophotometer.
Pag. 4. Lines 182-186. Add name, model, and country of origin of centrifuges and freeze dryer.
Results and discussion
All figures. Improve the sharpness of all figures. The results cannot be seen.
Pag. 6. Lines 244-275. Only figures 1H and 1D are cited in the text. What about the rest of the figures? I recommend that if they are not described or discussed, they should be removed.
Pag. 7-8. Lines 291-275. Figure 2E needs to be cited in text.
Pag. 10. Line 373. What is the figure number?
Pag. 11. Line 424. Figure 4A needs to be cited in text.
References
Review the information in reference 19 carefully.
Add the year of references 12, 18, 22, 35, 37 and 39.
Reviewer 3 Report
Comments and Suggestions for Authors
An interesting study has been done on the use of Lactobacillus fermentum to improve the sour taste of sea buckthorn fruit vinegar. However, the manuscript presented is confusing and with too many mistakes.
Some remarks are as follows:
- The title should be checked
- The name of all species should be in italics throughout the text.
- page 60 intestinal flora should be changed to intestinal microbiota.
- page 111 Bacillus pasteurus aceticus ?
- page 131 ‘the juice was sterilised at 80 ºC for 30 min’ this is not possible, the spores are not destroyed at this temperature.
- p. 132 ‘Lactobacillus fermentum was then inoculated with Lactobacillus fermentans’?
- p. 137 Bartonella acetogenes ?
- p. 141 Saccharomyces cere visite?
- p. 164 Acinetobacter pasteurianus ?
-p. 215 between species or tissues?
-p 239 check
- Figure 1 Acetobacter baumannii?
- and others
The figures need to be improved a lot, because they are not clear. This study also requires sensory analysis and tasting of the products obtained.
The English could be improved to more clearly express the research.